# Air quality and obesity at older ages in China: The role of duration, severity and pollutants

Nan Zhang[1,2]*, Lei Wang[3,4]*, Min Zhang[5], James Nazroo[2,6]

**1** Social Statistics, Cathie Marsh Institute for Social Research (CMI), School of Social Sciences, The University of Manchester, Manchester, United Kingdom, **2** Manchester Institute for Collaborative Research on Ageing (MICRA), School of Social Sciences, The University of Manchester, Manchester, United Kingdom, **3** Key Laboratory of Watershed Geographic Sciences, Nanjing Institute of Geography and Limnology, Chinese Academy of Sciences, Nanjing, China, **4** Manchester Urban Institute, Urban Planning, School of Environment, Education and Development, The University of Manchester, United Kingdom, **5** Faculty of Education, University of Cambridge, Cambridge, United Kingdom, **6** Sociology, School of Social Sciences, The University of Manchester, Manchester, United Kingdom

\* nan.zhang-2@manchester.ac.uk(NZ); wanglei@niglas.ac.cn(LW)

## Abstract

### Background

Population ageing and air pollution have emerged as two of the most pressing challenges in China. However, little evidence has explored the impact of air pollution on obesity among older adults in China.

### Methods

The China Health and Retirement Longitudinal Study—a nationally representative sample of middle-aged and older Chinese was linked to the air pollution data at the city level. Multi-level logistic models were fitted on obesity status among older people in relation to different air quality measures such as chronic exposures to severities of air pollution and pollutants.

### Results

Air pollution was positively associated with increased risks of general obesity and abdominal obesity among older adults (N = 4,364) especially for those with disability. The marginal effects of average air quality index (AQI) on obesity suggest that one standard deviation increase in AQI is associated with increased risks of central obesity by 2.8% (95%CI 1.7% 3.9%) and abdominal obesity by 6.2% (95%CI 4.4% 8.0%). The risk of chronic exposures to light (and moderate), heavy and severe pollution on obesity elevated in a graded fashion in line with the level of pollution. Durations of exposure to PM2.5 and PM10 were significantly associated with increased risk of obesity among older people in China.

### Conclusions

Chronic exposures to severe air pollution and certain pollutants such as PM2.5 and PM10 raise the risk of obesity among older people in China and the relationships were stronger for

**Data Availability Statement:** This study used data from the China Health and Retirement Longitudinal Study (CHARLS), which can be accessed through https://g2aging.org/?section=downloads or http://charls.pku.edu.cn/pages/data/111/zh-cn.html. The

air quality data can be retrieved from the real-time data of China National Environmental Monitoring Center (http://www.cnemc.cn) as well as Qingyue Open Data Center (http://data.epmap.org/eia/air). We obtained air quality datasets by provinces for major cities from 2013 onwards, which contained all air quality measures used in this study. The air quality data can be shared publicly. We confirm the authors of this study had no special access privilege to the data other researchers would not have.

**Funding:** Nan Zhang received ESRC Global Challenge Research Fund Fellowship (ES/P009824/1) and Cathie Marsh Institute (CMI) seedcorn funding from University of Manchester. The funders have no role in study design, data collection and analysis, decision to publish, or preparation of the manuscript.

**Competing interests:** The authors have declared that no competing interests exist.

those with disability. Future policies that target these factors might provide a promising way of enhancing the physical health of older people.

## Introduction

Air pollution affects all regions of the world and residents in low-income cities are among those who suffer the highest levels of exposure. Worldwide, 91% of the world's population lives in places where air quality exceeds WHO guideline limits [1]. In low- and middle- income countries, 97% of cities with more than 100,000 inhabitants failed to meet the WHO air quality guidelines [2]. In the world's most populous and highly polluted economies such as China and India, air pollution levels have increased exponentially and therefore curbing air pollution has risen to become a national priority [3, 4]. By the end of 2017, only 30% out of 365 major Chinese cities at prefectural level or higher successfully met the World Health Organization's recommended annual average PM2.5 level of $35\mu g/m^3$ [5]. Air pollution has been found to be adversely associated with health status. Pollutants with the strongest evidence for public health concern, include particulate matter (PM), ozone (O3), carbon monoxide (CO), nitrogen dioxide (NO2) and sulphur dioxide (SO2) [6, 7].

Ambient (outdoor) air pollution is one of the major causes of death and disease having contributed 7.6% to all deaths in 2016 [8]. An estimated 4.2 million premature deaths globally are linked to ambient air pollution, mainly from heart disease, stroke, chronic obstructive pulmonary disease, lung cancer, and acute respiratory infections, whereas 50% of all deaths are due to cardiovascular disease [1]. Elevated air pollution levels are associated with increased cardiovascular risks such as stroke, hypertension, diabetes, and obesity [9–14]. Severe air pollution may strongly discourage residents to take physical activities and in turn increases the risks of weight gain. The associations between air pollution and body weight status may vary by age groups, gender and types of air pollutants. Several pathways have been proposed to link air pollution and obesity directly and indirectly, through metabolic dysfunction [15], increased oxidative stress and adipose tissue inflammation [16, 17], onset of chronic disease [18], and disruption of regular physical activity [19]. Animal research in China has demonstrated that, compared to filtered air, chronic exposure to highly polluted air in Beijing increased the risk of obesity and metabolic dysfunction [20]. In addition, epidemiological evidence suggests that obesity may exacerbate susceptibility to the impacts of air pollution on cardiovascular health [13] and stroke among older adults in particular among women [12]. Physical inactivity is a strong predictor of the obesity epidemic, and it can be promoted or hindered by environmental pollution. Evidence shows that environmental quality plays an important part in the association between physical inactivity and obesity [21]. Previous longitudinal evidence has suggested that air pollution measured by ambient PM2.5 significantly deterred old residents in Beijing from engaging in daily activity and induced longer hours of inactivity such as nighttime/daytime sleeping [22]. It is important to examine if physical activity may mediate the relationship between air quality and obesity risks.

Previous studies on the impact of air pollution on health among the elderly in China are not without limitations. Some studies [23, 24] used either self-rated health or self-reported air quality measures that were prone to recall error and social desirability bias [25]. Longitudinal evidence of a 11-year follow-up found that after adjusting for provincial- and individual-level social and economic characteristics, long-term exposure to air pollution $(SO^2)$ is a significant risk factor for all-cause mortality among older Chinese adults over 65 years old [26]. However, air pollution indicators were measured at the provincial level, which may bias the results

because air quality conditions may vary substantially within each province. Using air pollution indicators at smaller geographical scales such as city can be helpful to yield less biased results. Also, it is important to examine the chronic exposure to air pollution because the negative effect of exposure to air pollution on health among older people may accumulate over time, as proposed by the accumulation hypothesis [27].

Current evidence regarding the impact of air pollution on body weight status among older people remains mixed. It may depend on the duration of exposure to air pollution and levels of air pollution. China is facing two of the most pressing challenges—population ageing and air pollution that may threaten China's sustainable development. However, little evidence has been in place to explore the influence of air pollution on health and wellbeing of the vulnerable elderly in China. The aim of the present study is to address the previous limitations in the literature and to investigate the associations between air pollution (duration, severity and different pollutants) and risk of obesity among older Chinese residents. Moreover, this study further examines whether physical activity mediates the relationship between air pollution and obesity among the Chinese elderly.

## Materials and methods

### Data

This study used data from China Health and Retirement Longitudinal Study (CHARLS). The CHARLS aims to collect biannually a nationally representative sample of Chinese residents aged over 45. The baseline national wave of CHARLS was fielded in 2011 and included about 10,000 households and 17,500 individuals across 150 counties/districts and 450 villages/resident committees in 28 provinces. The CHARLS adopts multi-stage stratified probability proportional to scale (PPS) sampling. The respondents were interviewed face-to-face in their homes via the computer-assisted personal interviewing (CAPI) technology. We used the newly released fourth sweep of the CHARLS released in 2015. The Ethical Review Committee of Peking University granted ethical approval for the study. All the participants provided signed, informed consent at the time of participation. Informed consent was obtained from all individual participants.

### Sample

Of 19,397 sample members from the CHARLS, an unmatched sample of 1,598 was excluded when linking to air quality data and local GDP per capita. Using the cut-point of 60 years for older adults, we restricted our sample to older adults who were aged 60 and above. Among 17,799 matched participants, 8,703 who were younger than 60 were dropped from the analytic sample. We then removed those who had missing data on obesity outcomes (n = 1,787) and yielded a sample of 7,309. Among all of participant households, only a random subsample (half households) was asked questions related to physical activity. The final analytical samples comprised 4,364 complete cases of older adults (on outcomes and covariates except for physical activity).

### Outcome

Anthropometric data such as the respondents' standing height, weight and waist circumference (WC) were measured by the trained interviewers with standardized equipment. Body mass index (BMI) and waist circumferences (WC) were used to define the obesity status of respondents. BMI ($kg/m^2$) was calculated as body weights (kg) divided by height squared ($m^2$); Chinese men and women with a BMI larger than 28 would be regarded as obese [28]. WC

measures abdominal obesity. A WC $\geq$ 90 cm for men and a WC $\geq$ 80 cm for women denote abdominal obesity in China [29]. BMI (and general obesity) is easy to measure and remains the most widely used tool to screen obesity risk among population. However, emerging clinical and epidemiological evidence suggests that abdominal obesity measures, such as WC, perform well in predicting the central distribution of adiposity which show strong associations with CVD risks [30–32], particularly among Chinese elderly [33]. Moreover, ageing is associated with considerable changes in body composition such as muscle and height loss and gain in fat mass [34, 35]. Thus, BMI may not be the most reliable indicator of obesity among older people. In this study, both general obesity and abdominal obesity were used to cross-validate our estimates.

## Air quality index

Air quality index (AQI) has been used worldwide to inform the public of levels of air pollution and associated health risks. China's Ministry of Environmental Protection (MEP) is responsible for measuring the level of air pollution in China. Air quality data were drawn on from Environment Quality Report of major cities in 2015 across China recorded by The Ministry of Ecology and Environment. Air quality data of 118 cities were employed in the analyses.

The annual average AQI is calculated based on the daily concentrations of the six criteria pollutants, namely sulfur dioxide ($SO_2$), nitrogen dioxide ($NO_2$), suspended particulates smaller than 10 μm in aerodynamic diameter (PM10), suspended particulates smaller than 2.5 μm in aerodynamic diameter (PM2.5) [7] carbon monoxide (CO), and ozone ($O_3$) measured at the monitoring stations across each city [36]. The average AQI ranges 0–500. Air pollution poses threats to physical health to varying extents [36]: (1) AQI $\leq$ 100 are acceptable air quality in China; (2) AQI [101–150] indicate light pollution: Healthy people may experience slight irritations and sensitive groups will be slightly affected; (3) AQI [151–200] indicate moderate pollution: Sensitive individuals will experience more serious conditions; the hearts and respiratory systems of healthy people may be affected; (4) AQI [201–300] indicate heavy pollution: Healthy people will commonly show symptoms; people with respiratory or heart diseases will be significantly affected and will experience reduced endurance in activity; (5) AQI $\geq$ 301 indicate severe pollution: Healthy people will experience reduced endurance in activity and may also show noticeably strong symptoms; other illnesses may be triggered in healthy people; elders and the sick should remain indoors and avoid exercise; healthy individuals should avoid outdoor activity.

AQI was used as an indicator of general air pollution status. We use the standardized AQI by subtracting off the mean and dividing by the standard deviation to represent the air pollution levels in the regression analysis. We also evaluated the health effects of air pollution in relation to the duration of exposure including: (1) the number of days of exposure to different levels of air pollution including light and moderate pollution (AQI $\geq$ 101), heavy pollution (AQI $\geq$ 201) and severe pollution (AQI $\geq$ 301), respectively, and (2) the number of days of exposure to major pollutants each year because Environment Quality Report provided the most important pollutant every day among six pollutants including PM2.5, PM10, CO, $NO_2$, $SO_2$ and $O_3$.

## Covariates

Household consumption per capita at the household level that combines measures of consumption of food, non-durable non-food items, durable goods and housing was used to capture household economic status. Compared with income data, consumption has been found to be strongly associated with wellbeing [37] and become a preferred indicator to measure long-

run welfare levels [38]. Household consumption per capita was logarithmically transformed in the analyses.

Hukou (household registration system) status is important because it represents social origin of people in China that is associated with socioeconomic status and life chances [39]. People with an urban Hukou status are entitled to many exclusive benefits, including government-provided housing, healthcare, employment, and social security [40]. These are denied to people with a rural Hukou, even if they are migrant workers living in urban communities. Residence of living also matters. Urban residents usually benefit from better access to public infrastructure and economic opportunities compared to those living in rural areas.

We include the following individual characteristics, which are expected to vary across groups and to be relevant to physical health: gender (male and female), age, marital status (married or partnered, and other including separated, divorced, widowed or never married), and highest educational levels (illiteracy, primary and/or less than lower secondary school, and tertiary school).

In the CHARLS, respondents were asked to report whether they have any difficulty with the following activities of daily living (ADLs): dressing, bathing, eating, getting into or out of bed, toilet, controlling urination or defecation. Similarly, they report difficulties with instrumental ADLs (IADLs): household chores, cooking, shopping, managing money and taking medication. The number of difficulties with ADLs and IADLs were used as binary predictors (one or more, and no difficulties). Physical inactivity was a dichotomous variable with 0 indicating respondent taking part in vigorous or moderate physical activities for at least 10 minutes every week and 1 representing no or light activities. However, only a random subsample (half households) among all of participant households was asked about questions related to physical activity.

City-level controls were included. Local GDP per capital at the city level and manufacturing share of local GDP serving as a macro-environmental indicator that is correlated with air pollution and obesity have been controlled for. GDP per capita was obtained from National Bureau of Statistics of China and local government portals and was logarithmically transformed in the analyses, because of its skewed distribution. The number of hospital beds per 1000 population as a marker of the capacity of health care and provision at the city level was included. Manufacturing share of local GDP and the number of hospital beds per 1000 population were derived from China City Statistical Yearbook [41]. According to geographical and socio-economic differences Regional dummies were created: South west, North west, North east, Central, North, East, South. South west, which is the most economically deprived region in China, was set as the reference category. Regional dummies were controlled for to capture unobserved geographical variations in physical environment, public resources, economic development, and diet culture across China.

## Analytical strategy

We firstly geocoded at the city level the China Health and Retirement Longitudinal Study (CHARLS) and linked to the air quality data at the city level. We conducted a cross-sectional analysis based on CHARLS 2015. Multilevel models (individual at level 1, household at level 2, and community at level 3) were used to correct for non-independence of observations due to geographic clustering [42]. Air quality data of 118 cities were employed in the analyses. We did not take into account the city-level variance as it was small and non-significant. Wald test was used to test jointly statistical significance in multilevel logistic regressions for categorical outcomes including general obesity and abdominal obesity.

The models were fitted in the following sequence. Model 1 adjusted only for air pollution to examine the crude association between AQI and obesity. In Model 2, we fully controlled for

relevant socioeconomic, demographic covariates and disability. In order to examine if the association between air quality and obesity differ between elder adults with disability and without, Model 3 further adjusted for the interaction between AQI and disability based on Model 2. Following the Baron and Kenny's approach [43], we tested the mediating role of physical inactivity on the relationship between AQI and obesity measures: mediation may be present if there are significant relationships between air quality measures and physical activity as a mediator as well as between physical activity and obesity, along with attenuation of the main air quality measure-obesity relationship after adjusting for the mediator.

## Results

Table 1 summarizes socioeconomic and demographic characteristics, air quality, and obesity outcomes of older adults aged ≥ 60 from the CHARLS. We reported samples that were included into and those excluded from the final analytical sample. The general obesity rate among Chinese elders was 11.2% and abdominal obesity rate was 54.8%, which was consistent with other reports in China [44]. Among all older adult respondents, over 80% (84.33%) were married or partnered; one third (31.7%) were illiterate; 75.6% held a rural Hukou; and 60.4% were living in rural communities. A quarter (24.3%) of older adults had at least one difficulty in physical functioning. More than half of the older people (60.4%) engaged in physical activities. Regarding air quality measures, 101 days per year failed to meet the minimum Chinese air quality standard (AQI<100) and the major pollutants were PM2.5 (exposure averaged at 144 days per annum) and PM10 (exposure averaged at 76 days per annum). Table 1 shows that there are some important differences between the included and excluded sample, however, in many cases the differences are small in size. The exceptions to this are: gender, where the excluded sample has a greater proportion of women; marital status, where the excluded sample is more likely to be single; disability, where the excluded sample is more likely to have a limitation; physical activity, where the excluded sample is less likely to be inactive.

Tables 2 and 3 present multilevel regression results on standardized AQI for general obesity and abdominal obesity among older adults in China, respectively. Average AQI was positively associated with an increased risk of obesity in unadjusted models (OR = 1.427, 95%CI 1.275 1.596 for general obesity, and OR = 1.388, 95%CI 1.276 1.510) (Model 1 in Tables 2 and 3). After adjusting for relevant socioeconomic and demographic factors, average annual AQI was still positively associated with higher risks of general obesity (OR = 1.375, 95%CI 1.214 1.559) (Model 2 Table 2) and abdominal obesity (OR = 1.424, 95%CI 1.280 1.585) among older people in China (Model 2 Table 3). The marginal effects (based on Model 2) of average AQI on obesity suggest that one standard deviation increase in AQI is associated with increased risks of central obesity by 2.8% (95%CI 1.7% 3.9%) and abdominal obesity by 6.2% (95%CI 4.4% 8.0%) (S1 Table).

As shown in Model 3 (Tables 2 and 3), the interactions between AQI and disability were statistically significant for both general obesity (OR = 1.269, 95%CI 1.024 1.574) and abdominal obesity (OR = 1.275, 95%CI 1.058 1.538). This may indicate that the influences of air pollution on obesity of older adults in China were moderated by the level of physical functioning and differed between those with disability and those without disability. S1 Fig shows predicted probabilities of general obesity and abdominal obesity among older adults by disability: as air quality deteriorates, those elder adults with disability tend to be more vulnerable to air pollution than those without difficulties in daily activities.

Table 4 presents the multilevel regression results on days of exposures to various levels of severity of air pollution after adjusting for relevant factors. The severity of air pollution was significantly associated with increased risk of obesity among older people in a graded fashion.

**Table 1. Sample description of included and excluded older adults from the China Health and Retirement Longitudinal Study 2015.**

| Variables | Included sample N | Included sample Mean (sd)/Percentage | Excluded sample N | Excluded sample Mean (sd)/Percentage | Difference P-value |
|---|---|---|---|---|---|
| **Outcomes** | | | | | |
| General obesity | 4364 | 11.16% | 2945 | 10.12% | 0.159 |
| Abdominal obesity | 4359 | 54.85% | 2937 | 51.75% | <0.01 |
| **Age** (years) | 4364 | 67.53 (6.11) | 2929 | 69.32 (7.50) | <0.001 |
| **Gender** | | | | | <0.001 |
| Male | 2253 | 51.63% | 1366 | 46.38% | |
| Female | 2111 | 48.37% | 1579 | 53.62% | |
| **Marital status** | | | | | <0.001 |
| Not married or partnered | 684 | 15.67% | 806 | 27.38% | |
| Married or partnered | 3680 | 84.33% | 2138 | 72.62% | |
| **Education** | | | | | <0.001 |
| Illiterate | 1382 | 31.67% | 1154 | 39.24% | |
| Primary or secondary | 2654 | 60.82% | 1632 | 55.49% | |
| Tertiary | 328 | 7.52% | 155 | 5.27% | |
| **Household consumption per capita** (Logged) (RMB) | 4364 | 8.97 (0.99) | 341 | 8.94 (1.04) | 0.622 |
| **Hukou** | | | | | <0.001 |
| Urban Hukou | 1065 | 24.40% | 432 | 17.50% | |
| Rural Hukou | 3299 | 75.60% | 2037 | 82.50% | |
| **Place of residence** (Ref: Urban) | | | | | <0.05 |
| Urban communities | 1729 | 39.62% | 1098 | 37.28% | |
| Rural communities | 2635 | 60.38% | 1847 | 62.72% | |
| **Disability** | | | | | <0.001 |
| No difficulty | 3303 | 75.69% | 2064 | 70.49% | |
| At least one difficulty | 1061 | 24.31% | 864 | 29.51% | |
| **Physical inactivity** | | | | | <0.001 |
| Active | 1282 | 60.44% | 738 | 54.71% | |
| Inactive | 839 | 39.56% | 611 | 45.29% | |
| **GDP per capita** (Logged) (RMB) | 4364 | 10.58 (0.53) | 2945 | 10.62 (0.54) | <0.01 |
| **Manufacturing share of GDP, %** | 4364 | 45.99 (7.85) | 2945 | 45.81 (7.72) | 0.337 |
| **Number of hospital beds per 1000** | 4364 | 4.67 (1.50) | 2945 | 4.69 (1.65) | 0.461 |
| **Air quality** | | | | | |
| Average AQI | 4364 | 88.73(23.14) | 2945 | 85.72(23.97) | <0.001 |
| No. of days AQI $\geq$ 101 | 4364 | 101(64) | 2945 | 94(66) | <0.001 |
| No. of days AQI $\geq$ 201 | 4364 | 15(16) | 2945 | 13(16) | <0.001 |
| No. of days AQI $\geq$ 301 | 4364 | 3(6) | 2945 | 3(5) | <0.01 |
| ***No. of days exposed to*** | | | | | |
| SO2 | 4364 | 4(10) | 2945 | 4(10) | 0.570 |
| NO2 | 4364 | 9(16) | 2945 | 11(20) | <0.001 |
| PM2.5 | 4364 | 144(60) | 2945 | 136(58) | <0.001 |
| PM10 | 4364 | 76(41) | 2945 | 76(41) | <0.001 |
| O3 | 4364 | 71(39) | 2945 | 71(39) | 0.768 |
| CO | 4364 | 3(8) | 2945 | 3(8) | <0.01 |

AQI, Air quality index; GDP, Gross Domestic Product

**Table 2. Regression results (Odds ratio and 95%CI) for the association between air pollution and general obesity among older people from the China Health and Retirement Longitudinal Study 2015.**

| Variables | Model 1[a] | | Model 2[b] | | Model 3[c] | |
|---|---|---|---|---|---|---|
| | OR | 95%CI | OR | 95%CI | OR | 95%CI |
| Intercept | 0.101*** | (0.080, 0.127) | 0.012* | (0.0004, 0.388) | 0.013* | (0.0004, 0.406) |
| Average AQI (Standardized) | 1.427*** | (1.275, 1.596) | 1.375*** | (1.214, 1.559) | 1.272*** | (1.103, 1.468) |
| Disability (Ref: No difficulty) | | | 1.840*** | (1.454, 2.329) | 1.712*** | (1.338, 2.190) |
| Average AQI*Disability | | | | | 1.269* | (1.024, 1.574) |
| Age (Centred at 60 years) | | | 0.955*** | (0.936, 0.974) | 0.955*** | (0.936, 0.974) |
| Sex (Ref: Male) | | | 1.659*** | (1.322, 2.081) | 1.658*** | (1.322, 2.080) |
| Marital status (Ref: No) | | | | | | |
| Married or partnered | | | 1.125 | (0.837, 1.530) | 1.121 | (0.824, 1.525) |
| Education (Ref: Illiterate) | | | | | | |
| Primary or secondary | | | 0.781 | (0.606, 1.005) | 0.784 | (0.609, 1.010) |
| Tertiary | | | 0.544* | (0.332, 0.890) | 0.549* | (0.335, 0.898) |
| Household consumption per capita (Logged) | | | 1.056 | (0.940, 1.187) | 1.060 | (0.944, 1.191) |
| *Hukou* (Ref: Urban) | | | 0.730* | (0.534, 0.997) | 0.735 | (0.538, 1.003) |
| Place of residence (Ref: Urban) | | | 0.747* | (0.563 0.992) | 0.746* | (0.562, 0.990) |
| GDP per capita (Logged) | | | 1.148 | (0.775, 1.70) | 1.144 | (0.773, 1.692) |
| Manufacturing share of GDP, % | | | 0.997 | (0.978, 1.015) | 0.996 | (0.978, 1.015) |
| Number of hospital beds per 1000 | | | 1.056 | (0.934, 1.194) | 1.056 | (0.934, 1.193) |
| N | 4364 | | 4364 | | 4364 | |

AQI, Air quality index; GDP, Gross Domestic Product; OR, odds ratio; CI, confidence interval

[a] Model 1 adjusted for average AQI

[b] Model 2 further adjusted for age, gender, education, log transformation of household consumption per capita, *Hukou*, place of residence, disability, log transformation of local GDP, manufacturing share of GDP, number of hospital beds per 1000 and region (not reported)

[c] Model 3 further adjusted for the interaction between average AQI and disability based on Model 2

*$p < 0.05$

** $p < 0.01$

***$p < 0.001$

The odds ratios of chronic exposures to light and moderate pollution, heavy pollution and severe pollution on general obesity were 1.005 (95%CI 1.003 1.007), 1.018 (95%CI 1.011 1.026) and 1.048 (95%CI 1.029 1.067), respectively. The odds ratios for abdominal obesity are 1.005 (95%CI 1.004 1.007), 1.021% (95%CI 1.014 1.027) and 1.051 (95%CI 1.032 01.070), respectively. Among six pollutants, chronic exposure to PM2.5 and PM10 appear to be significantly associated with increased risk of obesity among older people in China (Table 5).

We further adjusted for physical activity based on Model 2 from Tables 2 and 3 (S2 Table) to examine if physical activity mediated the relationship between air pollution and obesity. Following the Baron and Kenny's approach [43], we tested the relationships between air quality measures and physical activity (S3 Table), as well as between physical activity and obesity. Physical inactivity may mediate the impacts of AQI on obesity if it was found to be significantly associated with both AQI and obesity status and if the incorporation of physical activities substantially altered the associations between AQI and obesity. As the fully adjusted models did not find significant relationships between physical inactivity and obesity status (S2 Table), we did not find evidence that physical activity meaningfully mediates the relationship between air quality and obesity among older adults in China. However, it is important to note that the interpretation should be regarded with caution as physical activity was only asked from a subsample of eligible older adults.

**Table 3. Regression results (Odds ratio and 95%CI) for the association between air pollution and abdominal obesity among older people from the China Health and Retirement Longitudinal Study 2015.**

| Variables | Model 1[a] | | Model 2[b] | | Model 3[c] | |
|---|---|---|---|---|---|---|
| | OR | 95%CI | OR | 95%CI | OR | 95%CI |
| **Intercept** | 1.246*** | (1.147, 1.354) | 1.384 | (0.101, 19.042) | 1.406 | (0.101, 19.491) |
| **Average AQI** (Standardized) | 1.388*** | (1.276, 1.510) | 1.424*** | (1.280, 1.585) | 1.349*** | (1.203, 1.511) |
| **Disability** (Ref: No difficulty) | | | 1.597*** | (1.331, 1.917) | 1.607*** | (1.337 1.932) |
| **Average AQI*Disability** | | | | | 1.275* | (1.058, 1.538) |
| **Age** (Centred at 60 years) | | | 0.991 | (0.978, 1.004) | 0.991 | (0.978, 1.004) |
| **Sex** (Ref: Male) | | | 7.427*** | (6.007, 9.182) | 7.462*** | (6.030, 9.235) |
| **Marital status** (Ref: No) | | | | | | |
| Married or partnered | | | 1.224 | (0.989, 1.515) | 1.223 | (0.987, 1.515) |
| **Education** (Ref: Illiterate) | | | | | | |
| Primary or secondary | | | 1.043 | (0.864, 1.257) | 1.050 | (0.870, 1.267) |
| Tertiary | | | 1.307 | (0.928, 1.841) | 1.322 | (0.938, 1.863) |
| **Household consumption per capita** (Logged) | | | 1.085* | (1.001, 1.177) | 1.088* | (1.003, 1.180) |
| *Hukou* (Ref: Urban) | | | 0.686** | (0.543, 0.867) | 0.690** | (0.546, 0.872) |
| **Place of residence** (Ref: Urban) | | | 0.618*** | (0.496, 0.771) | 0.616*** | (0.494, 0.769) |
| **GDP per capita** (Logged) | | | 0.786 | (0.580, 1.066) | 0.784 | (0.578, 1.065) |
| **Manufacturing share of GDP, %** | | | 0.996 | (0.982, 1.010) | 0.996 | (0.981, 1.010) |
| **Number of hospital beds per 1000** | | | 1.20*** | (1.086, 1.326) | 1.201*** | (1.087, 1.328) |
| **N** | 4398 | | 4398 | | 4398 | |

AQI, Air quality index; GDP, Gross Domestic Product; OR, odds ratio; CI, confidence interval

[a] Model 1 only adjusted for AQI

[b] Model 2 further adjusted for age, gender, education, log transformation of household consumption per capita, *Hukou*, place of residence, disability, log transformation of local GDP, manufacturing share of GDP, number of hospital beds per 1000 and region (not reported)

[c] Model 3 further adjusted for the interaction between average AQI and disability based on Model 2

*p<0.05

** p<0.01

***p<0.001

## Discussion

Linking a nationally representative sample of older Chinese adults to air quality data at the city level, this study investigated the associations of air quality as well as chronic exposures to severities of air pollution and major pollutants with obesity (general obesity and abdominal obesity)

**Table 4. Regression results (odds ratio and 95%CI) for associations between chronic exposures to severity of air pollution among older people from the China Health and Retirement Longitudinal Study 2015.**

| No. of days exposed to | General obesity | | Abdominal obesity | |
|---|---|---|---|---|
| | OR | 95%CI | OR | 95%CI |
| **AQI> = 101** | 1.005*** | (1.003, 1.007) | 1.005*** | (1.004, 1.007) |
| **AQI> = 201** | 1.019*** | (1.011, 1.026) | 1.021*** | (1.014, 1.027) |
| **AQI> = 301** | 1.048*** | (1.029, 1.067) | 1.051*** | (1.032, 1.070) |

AQI, Air quality index; GDP, Gross Domestic Product; OR, odds ratio; CI, confidence interval

Fully adjusted pollutants, age, gender, education, log transformation of household consumption per capita, *Hukou*, place of residence, region, log transformation of local GDP per capita, manufacturing share of GDP, number of hospital beds per 1000 and disability (not reported here).

*p<0.05

** p<0.01

***p<0.001

**Table 5. Regression results (Odds ratio and 95%CI) for associations between six pollutants and obesity among older people from the China Health and Retirement Longitudinal Study2015.**

| No. of days exposed to | General obesity | | Abdominal obesity | |
|---|---|---|---|---|
| | OR | 95%CI | OR | 95%CI |
| SO2 | 0.994 | (0.979, 1.008) | 0.994 | (0.984, 1.005) |
| NO2 | 0.999 | (0.990, 1.008) | 1.002 | (0.996, 1.010) |
| PM2.5 | 1.005*** | (1.002, 1.008) | 1.005*** | (1.002, 1.007) |
| PM10 | 1.007*** | (1.003, 1.012) | 1.005** | (1.001, 1.008) |
| O3 | 1.0004 | (0.996, 1.004) | 1.002 | (0.999, 1.005) |
| CO | 0.986 | (0.969, 1.003) | 0.990 | (0.980, 1.003) |

GDP, Gross Domestic Product; OR, odds ratio; CI, confidence interval

Fully adjusted pollutants, age, gender, education, log transformation of household consumption per capita, *Hukou*, place of residence, region, log transformation of local GDP per capita, manufacturing share of GDP, number of hospital beds per 1000 and disability (not reported here).

*p<0.05

** p<0.01

***p<0.001

among older Chinese adults. This study further examined whether physical inactivity mediates the relationship between air pollution and obesity among Chinese elderly. After adjusting for relevant controls, air quality is found to be significantly associated with increased risk of obesity (general obesity and abdominal obesity) among older Chinese adults, and this relationship was stronger for elder adults with disability. Exposure to severe air pollution was significantly associated with increased risk of obesity among older people in a graded fashion. Among six pollutants, chronic exposure to PM2.5 and PM10 appear to play a crucial part in the associations we discovered in the present paper. This study has contributed to the current literature on health impact of air pollution on weight status in China in three aspects. First, this study distinguished the influences of chronic exposure to severities of air pollution levels and different pollutants on obesity outcomes. Second, this study used two objective measures—general obesity and abdominal obesity to cross-validate the findings. Third, this study further examined the mediating role of physical activity in the relationship between air pollution and obesity among older adults in China.

Previous studies on the impact of air pollution on health among older adults in China have several limitations. For example, some studies [23, 24] relied on self-rated health and self-reported air quality measures that are subject to recall error and social desirability bias [25]. Zhang et al. (2018) found that long-term exposure to air pollution (SO$_2$) is a significant risk factor for all-cause mortality among older Chinese adults over 65 years old [26]. However, air pollution measured at the provincial level essentially disregarded the substantial variation within the provinces, which may leads to biased results because some sources of air pollutants such as PM2.5 are often generated locally. In comparison, air quality measures at a small spatial scale such as city that we used in the current study may be a more effective measure in developing the understanding of the impact of local air quality on health.

One of the possible explanations for the significant association between chronic exposure to air pollution and elevated risks of obesity we observed is that the negative effect of exposure to air pollution on health among older people accumulates over time, as argued by the accumulation hypothesis [27]. Epidemiological evidence suggests that obesity may increase susceptibility to the cardiovascular health effects of PM2.5 [13]. This is consistent with previous studies on the influence of air pollution on obesity and CVD risks in China [11, 12] and in

other countries [9, 10, 13]. We have tested gender differences in the associations between AQI and obesity by adding the interaction term between sex and AQI. The absence of the significant interaction between sex and AQI suggests that air pollution may have equal obesity impacts on women and men. Also, we found that chronic exposures to PM2.5 and PM10 play a significant part in the adverse impact of air pollution on obesity outcomes among older Chinese adults. On one hand, air pollution, such as PM, are known to increase infiltration and activation of immune-competent cells (i.e., monocytes and macrophages) in adipose and other tissues [45]. On the other, epidemiological evidence suggests that obesity may increase susceptibility to the impacts of PM2.5 on the cardiovascular conditions [13]. For example, obesity may enhance the effects of air pollution on the CVDs and stroke among Chinese adults in particular among women [12]. We did not find significant impacts of other pollutants such as CO, $NO_2$, $SO_2$ and $O_3$ on obesity outcomes at older ages. One of the possible explanations may be due to the fact that the durations of exposure to these pollutants are shorter than PM in China. Our estimates thus may be underpowered. Moreover, there is little evidence suggesting whether and how certain pollutants such as $NO_2$, $SO_2$ and $O_3$ may be held accountable for obesity [46] and it remains unknown for older people.

Obesity is an important risk factor for CVD that is prevalent among older people [47] and is one of the leading causes of death in China. CVD risk-related factors (i.e., obesity, hypertension) have been becoming increasingly common among the Chinese elderly [48–50] in the past few decades. Population ageing is estimated to bear two thirds of the total disease burden in China by 2030 [51], which will place a large financial burden on its healthcare and social welfare system. Reducing severe air pollution especially PM2.5 and PM10 may provide a promising way of improving obesity status of older people in China.

It is acknowledged that environmental quality modifies the association between physical inactivity and obesity [21]. Longitudinal evidence in Beijing has suggested that air pollution measured by ambient PM2.5 significantly discouraged older adults from engaging in daily activity and induced longer hours of inactivity such as nighttime/daytime sleeping. Evidence from longitudinal studies has suggested that the elderly tended to avoid taking outdoor physical activities due to severe PM2.5 pollution and in turn spent much time being inactive and lacked exercises that are beneficial to their health [22]. However, it mainly focused on a single highly urbanized and polluted city, which may not generalize the findings to other regions across China. In contrast, our study drew on nation-wide data from 28 provinces and nearly 120 cities across China and have better generalizability to nationwide. Our findings indicate that physical activity patterns did not appear to meaningfully mediate the detrimental impacts of air pollution on obesity outcomes among older residents in China. However, the result should be interpreted with caution as the interview questions related to physical activity were only asked from a subsample of eligible respondents. In addition, the oversimplified measure of physical activities that we used in this study may not adequately capture the duration and intensity of weight-related activity. Regarding functional disability, we discovered a significant interaction effect with AQI: unhealthy air quality exerted a stronger adverse impact on residents over 60 who experienced difficulties with at least one ADL or IADL as compared to those who considered themselves fully functional. Having disability may further restrict older people's mobility and physical activity, and, together with poor air quality, may increase the risk of obesity that is already alarmingly high among older people. It is important for future studies to further explore whether and to what extent different intensities of physical activity may help mitigate the negative impact of air pollution on the elderly obesity in China.

## Limitations

Several methodological limitations warrant cautious interpretation of our findings. First, we used the AQI as an average measure of air quality, which is based on the highest value among six individual pollutants, and did not consider the potential combined health effects of exposure to multiple pollutants [52]. Also, we were unable to further examine concentrations of air pollutants as they were not available from the datasets. Second, this study aimed to elucidate the behavioral pathway through which air pollution might operate on obesity outcomes among older people. However, physical activity was measured only as a dichotomous variable that may not adequately describe the duration and intensity of weight-related activity. The third limitation is related to the bias of unobserved variables. Despite that we have controlled for as many relevant covariates as the data allows we could not take into account risk factors that are not available in the CHARLS data but are potentially relevant to obesity status such as nutrition, diet, and environmental factors other than air quality such as water, land and built environment [21]. Although the CHARLS is a longitudinal study, to our knowledge, there is no consistent annual measure for air quality available in China. As a compromise we conducted a cross-sectional analysis using multilevel modeling that controls for the unobserved heterogeneity at the household and community levels. Early-life exposure to ambient particles has been found to lead to either stronger susceptibility to diet-induced weight gain in adulthood or higher insulin resistance, adiposity, and inflammation [20]. Therefore a life-course approach can be fruitful in exploring the complex association between air quality and weight gain. Moreover, our analytical sample consisted of 4364 out of 7309 eligible participants. There are some important differences between the included and excluded sample that may bias our estimates and may comprise the generalizability of our findings. However, in many cases the differences are small in size. The exceptions to this are: gender, where the excluded sample has a greater proportion of women; marital status, where the excluded sample is more likely to be single; disability, where the excluded sample is more likely to have a limitation; physical activity, where the excluded sample is less likely to be inactive. Importantly, these factors are all included in our models, which gives more confidence in the conclusions drawn. Future studies may consider biomarkers of obesity related to glucose-insulin homeostasis, adipose tissue inflammatory biomarkers to further strengthen causal inference [53]. Last but not least, it is important to acknowledge the regional differences in air pollution across China. It is important to further explore how regional variations play a part in shaping the relationships between air quality and obesity among older people in China. This may provide robust evidence for policy-making process to specifically target more troublesome areas/regions, such as central and northern parts of China where obesity rates tend to be higher than other areas.

## Conclusions

Although we were unable to make definitive statement and causal inferences using the linked CHARLS and air quality data, our findings suggest that elevated air pollution is significantly associated with increased risk of general obesity and abdominal obesity among older adults in China especially among those with disability. The severity of air pollution was significantly associated with increased risk of obesity among older people in a graded fashion, whereas among six pollutants, chronic exposures to PM2.5 and PM10 play a strong role among older people in China. Future policies aiming to improve health and wellbeing of the elderly will benefit from tackling air pollution and reducing specific air pollutants.

## Supporting information

**S1 Table. Marginal impacts of average AQI on general obesity and central obesity among older people from the China Health and Retirement Longitudinal Study 2015 (based on Model 2).**
(DOCX)

**S2 Table. Regression results (Odds ratio and 95%CI) for testing mediating role of physical activity by further adjusting for physical activity.**
(DOCX)

**S3 Table. Regression results (Odds ratio and 95%CI) for associations between air quality measures and physical inactivity among older people from the China Health and Retirement Longitudinal Study 2015.**
(DOCX)

**S1 Fig.** Predicted general obesity (Left) and abdominal obesity (Right) among older adults by disability, the China Health and Retirement Longitudinal Study 2015.
(DOCX)

## Acknowledgments

We would like to acknowledge the team of the China Health and Retirement Longitudinal Study (CHARLS) 2015. We are grateful to all the participants who took part in the survey, and we also acknowledge all the subjects and households that participated in the survey for their cooperation. We would like to acknowledge China National Environmental Monitoring Center and Qingyue Open Data Center for air quality data.

## Author Contributions

**Conceptualization:** Nan Zhang, James Nazroo.

**Formal analysis:** Nan Zhang.

**Funding acquisition:** Nan Zhang.

**Investigation:** Nan Zhang, Lei Wang.

**Software:** Lei Wang.

**Validation:** Min Zhang.

**Visualization:** Lei Wang.

**Writing – original draft:** Nan Zhang.

**Writing – review & editing:** Nan Zhang, Lei Wang, Min Zhang, James Nazroo.

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
