## [Decision Letter · Decision Letter 0]

2 Oct 2019

PONE-D-19-21831

Exposure to ambient air pollution and obesity at older ages in China: the role of duration, severity and pollutants

PLOS ONE

Dear Dr Zhang,

Thank you for submitting your manuscript to PLOS ONE. After careful consideration, we feel that it has merit but does not fully meet PLOS ONE’s publication criteria as it currently stands. Therefore, we invite you to submit a revised version of the manuscript that addresses the points raised during the review process.

We would appreciate receiving your revised manuscript by Nov 16 2019 11:59PM. To enhance the reproducibility of your results, we recommend that if applicable you deposit your laboratory protocols in protocols.io, where a protocol can be assigned its own identifier (DOI) such that it can be cited independently in the future. For instructions see: http://journals.plos.org/plosone/s/submission-guidelines#loc-laboratory-protocols

We look forward to receiving your revised manuscript.

Kind regards,

A. Kofi Amegah, PhD

Academic Editor

PLOS ONE

**Journal Requirements:**

2. Thank you for stating the following financial disclosure: “No”   a) Please provide an amended Funding Statement that declares *all* the funding or sources of support received during this specific study (whether external or internal to your organization) as detailed online in our guide for authors at http://journals.plos.org/plosone/s/submit-now.     b)  Please state what role the funders took in the study.  If any authors received a salary from any of your funders, please state which authors and which funder. If the funders had no role, please state: "The funders had no role in study design, data collection and analysis, decision to publish, or preparation of the manuscript."

c) If the study was unfunded, please state "The author(s) received no specific funding for this work."

3. We note that  Figure(s) 2,3,& 4 in your submission contain [map/satellite] images which may be copyrighted. All PLOS content is published under the Creative Commons Attribution License (CC BY 4.0), which means that the manuscript, images, and Supporting Information files will be freely available online, and any third party is permitted to access, download, copy, distribute, and use these materials in any way, even commercially, with proper attribution. For these reasons, we cannot publish previously copyrighted maps or satellite images created using proprietary data, such as Google software (Google Maps, Street View, and Earth). For more information, see our copyright guidelines: http://journals.plos.org/plosone/s/licenses-and-copyright.

a) You may seek permission from the original copyright holder of Figure(s) [#] to publish the content specifically under the CC BY 4.0 license.  

**Comments to the Author**

1. Is the manuscript technically sound, and do the data support the conclusions?

Reviewer #1: Partly

Reviewer #2: Yes

2. Has the statistical analysis been performed appropriately and rigorously? 

Reviewer #1: No

Reviewer #2: Yes

3. Have the authors made all data underlying the findings in their manuscript fully available?

Reviewer #1: Yes

Reviewer #2: No

4. Is the manuscript presented in an intelligible fashion and written in standard English?

Reviewer #1: Yes

Reviewer #2: Yes

5. Review Comments to the Author

Reviewer #1: Reviewer comments for manuscript PONE-D-19-21831

The authors present findings from their study exploring the association of general and pollutant-specific air quality with general and central obesity, using data from CHARLS study. Although the manuscript is generally well-written, there are methodological issues which need to be addressed to improve the manuscript.

1. Adapt the title to reflect air quality index as the exposure metric

2. The study included 4364 out of 9069 eligible participants (48%). The authors should there present a table likely the current Table 1, where they summarize all the covariates between the included (4364) and the excluded (4732) CHARLS participants, and report the p-values of the differences in each covariate between the groups. You could then discuss the generalizability of This should replace the current Table 1 as it is not so informative on the research question.

3. Air quality indicators were assigned on city level. How many city-level clusters did the study include? How did you deal with clustering in AQI in the analyses? I suggest sensitivity analyses where city-level is considered as fixed as well as random effect, especially for the standardized continuous AQI models

4. It is unclear in the methodology, the definition of the overall AQI. This was mentioned in the limitations, but should be moved to the methodology. How was the standardization of the AQI done?

5. What is the correlation of physical inactivity and disability? I expect this to be very high, given that the disability should hinder physical activity, and the finding of a strong association between disability and physical inactivity showed in Table A3. I suggest doing the effect modification and mediation analysis on both physical activity and disability, limited to those who reported physical activity. If the correlation is very high, you may then consider combining both into one variable and do the mediation/effect modification in the entire 4364 sample.

6. Table A2: Include the AQI estimates from Tables 2 and 3 for direct comparison, instead of just referring to them in the title.

7. Consider reporting the effect estimates in 2 decimal places instead of 3. Looks cleaner.

8. Manuscript contains some typographical errors that should be corrected

Reviewer #2: This study looks at the association between air pollution (AQI) and obesity in a cohort of Chinese elderly using a cross-sectional design. The detailed cohort characteristics and geographic variety of the participants are major strengths of the study, but the reliance on AQI for exposure assessment is a weakness. Overall the paper is well-written, and obesity (and central obesity) is an interesting outcome to evaluate given the relative lack of evidence in the current literature. I suggest several additions to strengthen the paper.

1. Consider adding to discussion as to why associations with particulate matter but not the other pollutants were observed, e.g. are there different potential mechanisms involved?

2. Introduction is rather long and could be shorter and more focused. Please consider combining paragraphs 1 and first half of paragraph 2, and shortening other paragraphs.

3. It’s a bit unclear about how the individual pollutants are modeled in section 2.4. What do you mean exactly by “days of exposure to major pollutants?” Same approach as AQI?

4. Only a percentage of the entire cohort was included in the study. Are there significant differences in cohort characteristics between the excluded participants and included participants in the study?

Minor Comments

1. Please state population size in the abstract.

2. Clarify the cross-sectional nature of the study design in the methods.

3. Clarify “standardized AQI” in section 2.4.

4. Consider restating “older Chinese with disability” to “elderly Chinese with disability” when discussing vulnerable subpopulations. Older Chinese sounds like age is also a modifier.

6. PLOS authors have the option to publish the peer review history of their article (what does this mean?). If published, this will include your full peer review and any attached files.

Reviewer #1: No

Reviewer #2: No

---

## [Author Response · Author response to Decision Letter 0]

28 Oct 2019

Journal Requirements:

Thank you. We have formmated the manuscript according to the requirements. 

2. Thank you for stating the following financial disclosure: “No” a) Please provide an amended Funding Statement that declares *all* the funding or sources of support received during this specific study (whether external or internal to your organization) as detailed online in our guide for authors at http://journals.plos.org/plosone/s/submit-now. b) Please state what role the funders took in the study. If any authors received a salary from any of your funders, please state which authors and which funder. If the funders had no role, please state: "The funders had no role in study design, data collection and analysis, decision to publish, or preparation of the manuscript."

c) If the study was unfunded, please state "The author(s) received no specific funding for this work."

Thank you. We amended funding statements and included it within the cover letter. 

3. We note that Figure(s) 2,3,& 4 in your submission contain [map/satellite] images which may be copyrighted. All PLOS content is published under the Creative Commons Attribution License (CC BY 4.0), which means that the manuscript, images, and Supporting Information files will be freely available online, and any third party is permitted to access, download, copy, distribute, and use these materials in any way, even commercially, with proper attribution. For these reasons, we cannot publish previously copyrighted maps or satellite images created using proprietary data, such as Google software (Google Maps, Street View, and Earth). For more information, see our copyright guidelines: http://journals.plos.org/plosone/s/licenses-and-copyright.

Thank you very much. 

The figures 2-4 were produced by authors through the software of ArcMap 10.4. The shapefile of China’s municipal boundaries (polygons) are accessed from the datasets of the National Geomatics Center of China (http://www.ngcc.cn/ngcc/). Therefore, we own the copyrights. They are free to download by an application. We merged corresponding municipal boundaries to obtain seven regions of Southwest China, Northwest China, Northeast China, Central China, North China, East China, South China according to the regional division of China’s national official statistical yearbook. Consequently, we mapped the polygons of municipalities with different colors with the attributes their air quality index, exposures to PM2.5 and exposures to PM10 acquired from the real-time data of China National Environmental Monitoring Center (http://www.cnemc.cn). The website publishes daily air quality data and major pollutants of Chinese municipalities in the past years.

Reviewer #1: Reviewer comments for manuscript PONE-D-19-21831

The authors present findings from their study exploring the association of general and pollutant-specific air quality with general and central obesity, using data from CHARLS study. Although the manuscript is generally well-written, there are methodological issues which need to be addressed to improve the manuscript.

Thank you for your comments. We have addressed the methodological issues that you kindly raised and please see our responses and an updated version. 

1. Adapt the title to reflect air quality index as the exposure metric

Thank you. We have modifited the title ‘Air quality and obesity at older ages in China: The role of duration, severity and pollutants’.

2. The study included 4364 out of 9069 eligible participants (48%). The authors should there present a table likely the current Table 1, where they summarize all the covariates between the included (4364) and the excluded (4732) CHARLS participants, and report the p-values of the differences in each covariate between the groups. You could then discuss the generalizability of This should replace the current Table 1 as it is not so informative on the research question.

Thank you very much. We replaced the orginal table with the following table. We have updated results according to the updated table and further discussed the influences on our findings in the discussion. 

After removing those who had missing data on obesity outcomes (n=1,787), the eligible older adults dropped to 7,309. Our analysis was based on 4364 out of 7,309. The table below does show that there are some important differences between the included and excluded sample that may bias our estimates and may comprise the generalizability of our findings, however, in many cases the differences are small in size. The exceptions to this are: gender, where the excluded sample has a greater proportion of women; marital status, where the excluded sample is more likely to be single; disability, where the excluded sample is more likely to have a limitation; physical activity, where the excluded sample is less likely to be inactive. Importantly, these factors are all included in our models, which gives more confidence in the conclusions drawn. 

Table 1. Sample description of included and exluded older people from the China Health and Retirement Longitudinal Study 2015

 Included sample Excluded sample Difference

Variables N Mean (sd)/Percentage N Mean (sd)/Percentage P-value

Outcomes 

General obesity 4364 11.16% 2945 10.12% 0.159

Abdominal obesity 4359 54.85% 2937 51.75% <0.01

Age (years) 4364 67.53 (6.11) 2929 69.32 (7.50) <0.001

Gender <0.001

Male 2253 51.63% 1366 46.38% 

Female 2111 48.37% 1579 53.62% 

Marital status <0.001

Not married or partnered 684 15.67% 806 27.38% 

Married or partnered 3680 84.33% 2138 72.62% 

Education <0.001

Illiterate 1382 31.67% 1154 39.24% 

Primary or secondary 2654 60.82% 1632 55.49% 

Tertiary 328 7.52% 155 5.27% 

Household consumption per capita (Logged) (RMB) 4364 8.97 (0.99) 341 8.94 (1.04) 0.6220

Hukou <0.001

Urban Hukou 1065 24.40% 432 17.50% 

Rural Hukou 3299 75.60% 2037 82.50% 

Place of residence (Ref: Urban) <0.05

Urban communities 1729 39.62% 1098 37.28% 

Rural communities 2635 60.38% 1847 62.72% 

Disability <0.001

No difficulty 3303 75.69% 2064 70.49% 

At least one difficulty 1061 24.31% 864 29.51% 

Physical inactivity <0.001

Active 1282 60.44% 738 54.71% 

Inactive 839 39.56% 611 45.29% 

GDP per capita (Logged) (RMB) 4364 10.58 (0.53) 2945 10.62 (0.54) <0.01

Manufacturing share of GDP, % 4364 45.99 (7.85) 2945 45.81 (7.72) 0.3373

Number of hospital beds per 1000 4364 4.67 (1.50) 2945 4.69 (1.65) 0.4612

Air quality 

Average AQI 4364 88.73(23.14) 2945 85.72(23.97) <0.001

No. of days AQI ≥ 101 4364 101(64) 2945 94(66) <0.001

No. of days AQI ≥ 201 4364 15(16) 2945 13(16) <0.001

No. of days AQI ≥ 301 4364 3(6) 2945 3(5) <0.01

No. of days exposed to 

SO2 4364 4(10) 2945 4(10) 0.5704

NO2 4364 9(16) 2945 11(20) <0.001

PM2.5 4364 144(60) 2945 136(58) <0.001

PM10 4364 76(41) 2945 76(41) <0.001

O3 4364 71(39) 2945 71(39) 0.7684

CO 4364 3(8) 2945 3(8) <0.01

AQI, Air quality index; GDP, Gross Domestic Product 

3. Air quality indicators were assigned on city level. How many city-level clusters did the study include? How did you deal with clustering in AQI in the analyses? I suggest sensitivity analyses where city-level is considered as fixed as well as random effect, especially for the standardized continuous AQI models

Thank you very much. Air quality data of 118 cities were employed in the analyses. However, city-level variance was not taken into account, as it was small and non-significant compared to individual-, household- and community-levels. Moreover, we chose three-level instead of four-level (further adding city-level) multinomial logistic regressions because the latter failed to converage due to complex model structures. 

We have made changes to 2.6 Analytical strategy. 

4. It is unclear in the methodology, the definition of the overall AQI. This was mentioned in the limitations, but should be moved to the methodology. How was the standardization of the AQI done?

Thank you very much. We have modified the term ‘overall AQI’ to average AQI. 

The annual average AQI is calculated based on the daily concentrations of the six criteria pollutants, namely sulfur dioxide (SO2), nitrogen dioxide (NO2), suspended particulates smaller than 10 μm in aerodynamic diameter (PM10), suspended particulates smaller than 2.5 μm in aerodynamic diameter (PM2.5) (Ministry of Ecology and Environment of the People's Republic of China 2012a) carbon monoxide (CO), and ozone (O3) measured at the monitoring stations across each city (Ministry of Ecology and Environment of the People's Republic of China 2012b). The overall AQI ranges 0-500.

AQI was standardized by substracting off the mean and dividing by the standard deviation. We have made changes to the manuscript. 

5. What is the correlation of physical inactivity and disability? I expect this to be very high, given that the disability should hinder physical activity, and the finding of a strong association between disability and physical inactivity showed in Table A3. I suggest doing the effect modification and mediation analysis on both physical activity and disability, limited to those who reported physical activity. If the correlation is very high, you may then consider combining both into one variable and do the mediation/effect modification in the entire 4364 sample.

Thank you very much. 

We found correlation between physical inactivity and disability is 0.1329 in our study, which is quite low. We believe it might be partly due to how these two variables were measured in this study. Disability is an indicator of physical functioning which was based on activities of daily living (ADL) dressing, bathing, eating, getting into or out of bed, toilet, controlling urination or defecation, and instrumental ADLs (IADLs): household chores, cooking, shopping, managing money and taking medication. If participants have difficulty in one of the above activities, then he/she is considered living with disability. However, physical inactivity was a dichotomous variable with 0 indicating respondent taking part in vigorous or moderate physical activities for at least 10 minutes every week and 1 representing no or light activities. This is not an ideal measure as it was only asked for a subsample of eligible respondents and, more importantly, it is oversimplified that did not adequately capture the duration and intensity of weight-related activity. We have reported this in Discussion. 

In addition, we chose to do mediation analysis with physical inactivity because we think it is in the causal pathway from air pollution to obesity. The main reason for the interaction term with disability is because air pollution effect of obesity may be moderated by levels of physical functioning. 

6. Table A2: Include the AQI estimates from Tables 2 and 3 for direct comparison, instead of just referring to them in the title.

Thank you very much. We have modified the title of the Table A2. Since Tables 2 and 3 mainly focus on main effects of PM2.5 and obesity outcomes and the interaction with disability. The tables may seem confusing. Therefore, we decide to keep two tables separate. 

7. Consider reporting the effect estimates in 2 decimal places instead of 3. Looks cleaner.

Thank you very much. Because the magnitudes of some of our estimates are quite small and the changes in magnitudes may become invisible if we only report 2 decimials. Therefore, in order to show the precise effect size changes, we decided to keep 3 decimals. 

8. Manuscript contains some typographical errors that should be corrected

Reviewer #2: This study looks at the association between air pollution (AQI) and obesity in a cohort of Chinese elderly using a cross-sectional design. The detailed cohort characteristics and geographic variety of the participants are major strengths of the study, but the reliance on AQI for exposure assessment is a weakness. Overall the paper is well-written, and obesity (and central obesity) is an interesting outcome to evaluate given the relative lack of evidence in the current literature. I suggest several additions to strengthen the paper.

Thank you very much. AQI is an overall measure of air quality based on the six criteria pollutants, namely sulfur dioxide (SO2), nitrogen dioxide (NO2), suspended particulates smaller than 10 μm in aerodynamic diameter (PM10), suspended particulates smaller than 2.5 μm in aerodynamic diameter (PM2.5) carbon monoxide (CO), and ozone (O3). We have highlighted the limitation of using AQI in the discussion. 

1. Consider adding to discussion as to why associations with particulate matter but not the other pollutants were observed, e.g. are there different potential mechanisms involved?

Thank you very much. 

Also, we found that chronic exposures to PM2.5 and PM10 play a significant part in the adverse impact of air pollution on obesity outcomes among older Chinese adults. On one hand, air pollution, such as PM, are known to increase infiltration and activation of immune-competent cells (i.e., monocytes and macrophages) in adipose and other tissues (Zou 2010). On the other, epidemiological evidence suggests that obesity may increase susceptibility to the impacts of PM2.5 on the cardiovascular conditions (Weichenthal et al. 2014). For example, obesity may enhance the effects of air pollution on the CVDs and stroke among Chinese adults in particular among women (Qin et al. 2015). We did not find significant impacts of other pollutants such as CO, NO2, SO2 and O3 on obesity outcomes at older ages. One of the possible explanations may be due to the fact that the durations of exposure to these pollutants are shorter than PM in China. Our estimates thus may be underpowered. Moreover, there is little evidence suggesting whether and how certain pollutants such as NO2, SO2 and O3 may be held accountable for obesity (Dong et al. 2014) and it remains unknown for older people. 

2. Introduction is rather long and could be shorter and more focused. Please consider combining paragraphs 1 and first half of paragraph 2, and shortening other paragraphs.

Thank you very much. We have modified the introduction section and shorten some parts. Please see the updated version. 

3. It’s a bit unclear about how the individual pollutants are modeled in section 2.4. What do you mean exactly by “days of exposure to major pollutants?” Same approach as AQI?

Thank you very much. Yes, the approach was the same as AQI. 

Environment Quality Report provided the most important pollutant every day among six pollutants including PM2.5, PM10, CO, NO2, SO2 and O3, we calculate how many days of exposure to the major pollutants within one year. We have made changes and please see the updated version. 

4. Only a percentage of the entire cohort was included in the study. Are there significant differences in cohort characteristics between the excluded participants and included participants in the study?

Thank you very much. We replaced the orginal table with the new table and also reported the p-values of the differences in each covariate between the groups. Please refer to Table 1. We have updated results according to the updated table and further discussed the influences on our findings in the discussion. 

Minor Comments

1. Please state population size in the abstract.

Thank you, we have modified it. 

2. Clarify the cross-sectional nature of the study design in the methods.

Thank you, we have modified it. 

3. Clarify “standardized AQI” in section 2.4.

Thank you, we have modified it. AQI was standardized by substracting off the mean and dividing by the standard deviation.

4. Consider restating “older Chinese with disability” to “elderly Chinese with disability” when discussing vulnerable subpopulations. Older Chinese sounds like age is also a modifier.

Thank you, we have modified it. 

6. PLOS authors have the option to publish the peer review history of their article (what does this mean?). If published, this will include your full peer review and any attached files.   Do you want your identity to be public for this peer review? For information about this choice, including consent withdrawal, please see our Privacy Policy.

Reviewer #1: No

Reviewer #2: No

References 

Dong, G. H., Qian, Z., Liu, M. M., Wang, D., Ren, W. H., Flick, L. H., Fu, J., Wang, J., Chen, W., Simckes, M. and Trevathan, E. 2014. Ambient air pollution and the prevalence of obesity in Chinese children: The seven northeastern cities study. Obesity, 22, 795-800.

Ministry of Ecology and Environment of the People's Republic of China 2012a. Ambient Air Quality Standards "环境空气质量标准" Ministry of Ecology and Environment of the People's Republic of China.

Ministry of Ecology and Environment of the People's Republic of China 2012b. Technical Regulation on Ambient Air Quality Index (on trial) "环境空气质量指数（AQI) 技术规定（试行)”. Ministry of Ecology and Environment of the People's Republic of China.

Qin, X. D., Qian, Z., Vaughn, M. G., Trevathan, E., Emo, B., Paul, G., Ren, W. H., Hao, Y. T. and Dong, G. H. 2015. Gender-specific differences of interaction between obesity and air pollution on stroke and cardiovascular diseases in Chinese adults from a high pollution range area: A large population based cross sectional study. Sci Total Environ, 529, 243-8.

Weichenthal, S., Hoppin, J. A. and Reeves, F. 2014. Obesity and the cardiovascular health effects of fine particulate air pollution. Obesity, 22, 1580-9.

Zou, M. H. 2010. Is NAD (P) H oxidase a missing link for air pollution-enhanced obesity? Arteriosclerosis, Thrombosis, and Vascular Biology, 30, 2323-2324.

---

## [Decision Letter · Decision Letter 1]

25 Nov 2019

Air quality and obesity at older ages in China: the role of duration, severity and pollutants

PONE-D-19-21831R1

Dear Dr. Zhang,

We are pleased to inform you that your manuscript has been judged scientifically suitable for publication and will be formally accepted for publication once it complies with all outstanding technical requirements.

With kind regards,

A. Kofi Amegah, PhD

Academic Editor

PLOS ONE

Additional Editor Comments (optional):

Reviewers' comments:

Reviewer's Responses to Questions

**Comments to the Author**

1. If the authors have adequately addressed your comments raised in a previous round of review and you feel that this manuscript is now acceptable for publication, you may indicate that here to bypass the “Comments to the Author” section, enter your conflict of interest statement in the “Confidential to Editor” section, and submit your "Accept" recommendation.

Reviewer #1: All comments have been addressed

Reviewer #2: All comments have been addressed

2. Is the manuscript technically sound, and do the data support the conclusions?

Reviewer #1: Yes

Reviewer #2: Yes

3. Has the statistical analysis been performed appropriately and rigorously? 

Reviewer #1: Yes

Reviewer #2: Yes

4. Have the authors made all data underlying the findings in their manuscript fully available?

Reviewer #1: Yes

Reviewer #2: No

5. Is the manuscript presented in an intelligible fashion and written in standard English?

Reviewer #1: Yes

Reviewer #2: Yes

6. Review Comments to the Author

Reviewer #1: The authors have addressed all my comments and the manuscript has greatly improved from the original version. It should now be considered for publication.

Reviewer #2: All of my initial comments have been thoroughly addressed. I have no additional comments. Thank you.

7. PLOS authors have the option to publish the peer review history of their article (what does this mean?). If published, this will include your full peer review and any attached files.

Reviewer #1: No

Reviewer #2: No

---

## [Editor Report · Acceptance letter]

3 Dec 2019

PONE-D-19-21831R1 

Air quality and obesity at older ages in China: the role of duration, severity and pollutants 

Dear Dr. Zhang:

I am pleased to inform you that your manuscript has been deemed suitable for publication in PLOS ONE. Congratulations! Your manuscript is now with our production department. 

With kind regards,

on behalf of

Dr. A. Kofi Amegah 

Academic Editor

PLOS ONE